# Experimental Investigation of the Tension and Compression Creep Behavior of Alumina-Spinel Refractories at High Temperatures

**Lucas Teixeira [1,*]**, **Soheil Samadi [2]**, **Jean Gillibert [1]**, **Shengli Jin [2]**, **Thomas Sayet [1]**, **Dietmar Gruber [2]** and **Eric Blond [1]**

[1]    Université d'Orléans, Université de Tours, INSA-CVL, LaMé, 45072 Orléans, France; jean.gillibert@univ-orleans.fr (J.G.); thomas.sayet@univ-orleans.fr (T.S.); eric.blond@univ-orleans.fr (E.B.)

[2]    Chair of Ceramics, Montanuniversität Leoben, 8700 Leoben, Austria; soheil.samadi@unileoben.ac.at (S.S.); Shengli.Jin@unileoben.ac.at (S.J.); dietmar.gruber@unileoben.ac.at (D.G.)

*    Correspondence: lucas.breder-teixeira@univ-orleans.fr

**Abstract:** Refractory materials are subjected to thermomechanical loads during their working life, and consequent creep strain and stress relaxation are often observed. In this work, the asymmetric high temperature primary and secondary creep behavior of a material used in the working lining of steel ladles is characterized, using uniaxial tension and compression creep tests and an inverse identification procedure to calculate the parameters of a Norton-Bailey based law. The experimental creep curves are presented, as well as the curves resulting from the identified parameters, and a statistical analysis is made to evaluate the confidence of the results.

**Keywords:** refractories; creep; parameters identification

## 1. Introduction

Refractory materials, known for their physical and chemical stability, are used in high temperature processes in different industries, such as iron and steel making, cement and aerospace. These materials are exposed to thermomechanical loads, corrosion/erosion from solids, liquids and gases, gas diffusion, and mechanical abrasion [1]. In the steel industry, for equipment such as the steel ladle and BOF furnaces, the effect of creep strains and stress relaxation is of ultimate importance in the prediction of the lining performance.

Generally, the creep behavior of materials can be split in three stages. The first stage, called primary creep, presents a time-dependent strain rate which decreases with time. In the secondary creep stage, the strain rate is considered to be constant, and an approximate equilibrium between hardening and softening processes can be assumed [2]. Finally, in the third creep stage, the strain rate increases with time until the failure of the material [3].

The creep strain of a given material is highly dependent on the temperature and the applied stress or deformation [2]. For ceramic materials, including refractories, it has been demonstrated that the creep strain rate at one-dimensional tension load is considerably higher than the strain rate caused by a load with the same absolute value in compression [4]. It should be also noticed that some materials don't present all creep stages, sometimes going from primary to tertiary stage, or presenting only secondary and tertiary stages [5].

The creep models available in the literature are categorized into micromechanical and phenomenological models. Micromechanical models are used to evaluate what are the creep mechanisms taking place at a given material. The most common mechanisms that contribute for the

creep of ceramics are grain boundary sliding, diffusion and dislocation motion [5]. This methodology was applied to refractory materials by Martinez et al. [6]. The one-dimensional form of the most frequently used model for secondary creep strain rate equation in the context of micromechanical models is [7]:

$$\dot{\varepsilon} = \frac{KDGb}{kT} \left(\frac{b}{d}\right)^p \left(\frac{\sigma}{G}\right)^n \tag{1}$$

where $\sigma$ is the applied stress, $K$, $p$ and $n$ are material's constants, $G$ is the shear modulus, $b$ is the Burger's vector, $k$ is the Boltzmann's constant, $T$ is the temperature, $D$ is the diffusion coefficient and $d$ is the grain size.

Conversely, phenomenological models attempt to evaluate the effects of creep in a given material regardless of the possible mechanisms that could cause them. This normally results in simpler models with less parameters, at the cost of being less general. The most used phenomenological creep strain rate model is the Norton-Bailey's creep law. Its one-dimensional form is shown in Equation (2):

$$\dot{\varepsilon}_{cr} = A\sigma_{eq}^n \varepsilon_{cr}^a \tag{2}$$

where $\varepsilon_{cr}$ is the accumulated creep strain, $A$, $n$ and $a$ are temperature dependent material parameters and $\sigma_{eq}$ is the von Mises equivalent stress.

In the framework of nonlinear structural mechanics, phenomena involving permanent strains can be classified as time-independent and time-dependent plasticity (referred here as creep). The fundamental difference between them is that, in the later, the strain rate $\dot{\varepsilon}_{cr}$ can be explicitly defined as a function of the stress, while in the former this isn't possible due to mathematical constraints [8].

The function describing the creep strain rate can take many forms, depending on the material's behavior to be described. The Norton-Bailey law has been used to model the creep of refractories [3,4,9], and it seems to be appropriated also in the current work, as is shown in Section 5. For three-dimensional calculations, the Norton-Bailey equation is [10]:

$$\underline{\dot{\varepsilon}}_{cr} = \frac{3}{2} \frac{\underline{\underline{s}}}{\sigma_{eq}} A\sigma_{eq}^n \varepsilon_{eq}^a \tag{3}$$

where $\underline{s}$ is the deviatoric stress tensor, and $\varepsilon_{eq}$ is the equivalent creep strain tensor.

In the particular case of secondary creep, the parameter $a = 0$, and therefore the creep strain presents a linear curve.

In this work, the creep behavior of a shaped Alumina-Spinel refractory used as working lining material in steel ladles was characterized at 1300 °C using tension and compression tests. The main goal is to show how the scatter in the strain vs time curves can influence in the range of variation of the identified material parameters. This is done by plotting the confidence intervals for the curves based on a statistical procedure. Section 2 describes the material used for the mechanical tests. Section 3 shows the main characteristics of the mechanical tests used to obtain the creep curves. Section 4 presents the methodology used for the inverse identifications. Section 5 presents the experimental results and the identified parameters, as well as a discussion regarding the confidence interval of the data. Finally, in Section 6 the main conclusions of the work are presented.

## 2. Material

The Alumina-Spinel brick studied in this work has a maximum grain size of 3 mm, and it's used in the working lining of steel ladles in steel plants. According to the material's technical data sheet, it is mainly composed of 94% $Al_2O_3$, 5% $MgO$, 0.3% $SiO_2$ and 0.1% $Fe_2O_3$, with a bulk density of 3.13 g/cm$^3$ and apparent porosity of 19 vol%.

## 3. Experiments

The experimental creep curves were obtained using dedicated compression and tension testing machines. These machines were successfully used in previous works [3,9,11,12], and their main characteristics are presented in the next sections. The Alumina-Spinel material creep behavior was characterized at 1300 °C both in tension and in compression. For each load case, three different stress values were used, and for each stress value three different specimens were tested.

### 3.1. Uniaxial Compression Tests

To measure the displacements during the compression creep, two extensometers were positioned with a difference of 180° in relation to each other, and the distance between their corundum rods was 50 mm. The entire sample, the upper and the lower *SiC* pistons were located inside of a tubular furnace. A schematic representation of the compression creep experimental setup can be seen on Figure 1.

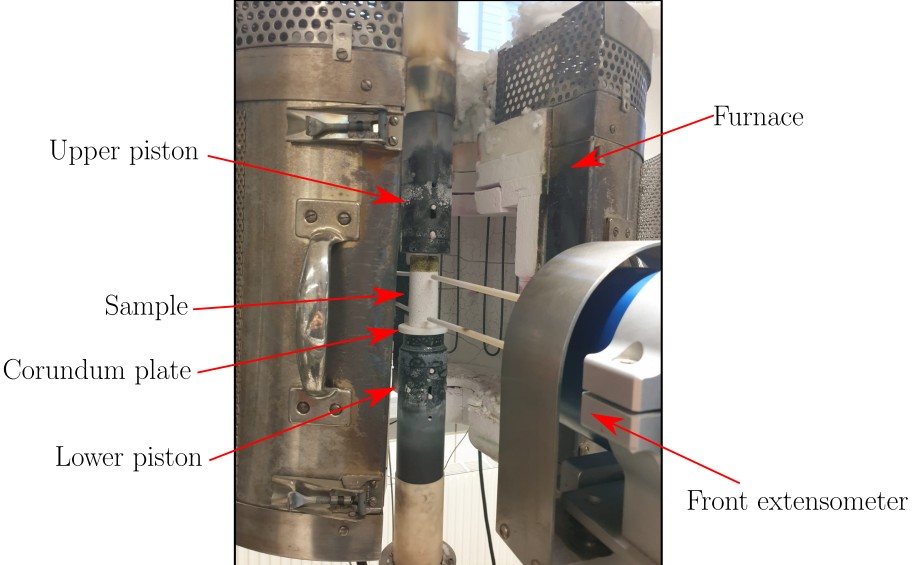

**Figure 1.** Compression creep experimental setup.

The samples used in the compression tests were drilled with 35 mm diameter and cut to a length of 70 mm, and a corundum plate was used at the contact between the sample and the lower piston to avoid chemical interactions.

At the beginning of the test a compressive pre-load of 50 N was applied to the sample to hold it in the correct position during the heating. The heating rate for the compression tests was 10 °C/min, and a 1h dwell was used to homogenize the sample's temperature.

The stress values used for the tests were 8 MPa, 9 MPa and 10 MPa.

### 3.2. Uniaxial Tensile Tests

For the tensile creep, two extensometers were used, with a distance between their corundum rods of 50 mm. To avoid damaging the sample during the gripping and consequent application of the load, the sample was glued to two water cooled adapters using a dedicated gluing device (Figure 2a). This device was designed to improve the alignment of the sample, avoiding the occurrence of bending loads [9]. The sample and the adapters were positioned at the testing machine and connected to water cooled grips, standing outside the furnace to avoid burning the glue. In this way, only a part of the sample stays inside the furnace. The tensile creep experimental setup is shown in Figure 2b.

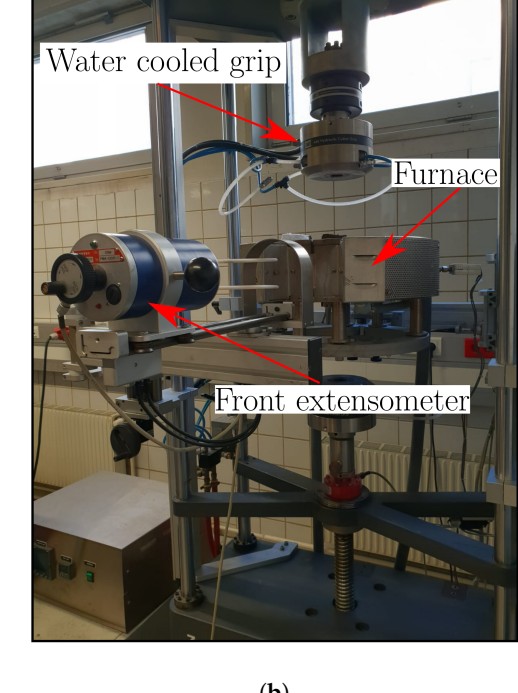

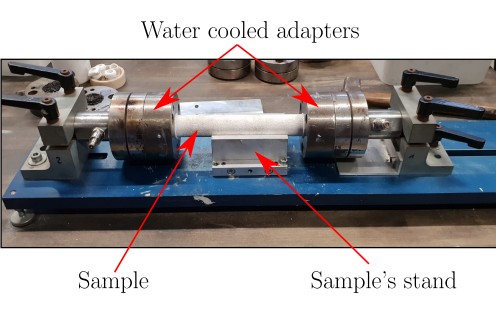

(**a**)                                                                                        (**b**)

**Figure 2.** Tensile creep experimental setup. (**a**) Gluing device. (**b**) Testing machine.

The samples used in the tensile tests were drilled with 30 mm diameter and cut to a length of 230 mm. A pre-load of 50 N was also applied in the tensile test, and the heating rate as 5 °C/min, with 1h dwell to reach the steady state temperature.

The stress values used for the tests were 0.15 MPa, 0.20 MPa and 0.25 MPa. Nevertheless, for the tensile load of 0.15 MPa the results of the three different tests presented a significant scatter, and they weren't considered during the calculations.

## 4. Methodology

### 4.1. Inverse Identification

To identify the material parameters from tension or compression creep tests, Equation (2) can be integrated using the trapezoidal rule [3], resulting in:

$$\varepsilon_{cr,i+1} \approx \left[ \varepsilon_{cr,i}^{1-a} + \frac{(1-a) \cdot A \cdot (\sigma_{i+1}^n + \sigma_i^n) \cdot (t_{i+1} - t_i)}{2} \right]^{\frac{1}{1-a}} \tag{4}$$

where $t$ is the time and $i$ is the time step index. In this way, depending on the stress variation over time and on a given set of material's parameters, it's possible to obtain an analytic calculation of the resulting time vs creep strain curve, which can later be compared to the experimental results using a least squares approach.

It should be taken into account that the experimental creep curves comprise both the elastic and creep strains ($\varepsilon_{tot}$), but Equation (2) doesn't account for the elastic part. Therefore, either the experimental data should be treated or the elastic strains should be included in the equation, following the expression for the one-dimensional case:

$$\varepsilon_{tot} = \varepsilon_{cr} + \frac{\sigma}{E} \tag{5}$$

where the second part of the right hand side of the equation corresponds to the elastic strain, being *E* the Young's modulus of the material.

The inverse identifications carried out in this work were done according to the following steps:

- **Step 1**: Definition of the input variables.

    1. Sample's diameter
    2. Young's modulus
    3. Type of creep (primary or secondary)
    4. Allowed range of variation for the material's properties
    5. Raw data from the tests (time-force-displacement tables)

- **Step 2**: Random definition of the initial guesses, depending on the variable's range of variation and the number of initial guesses.
- **Step 3**: For each of the initial guesses and each of the stress levels, calculate the analytic time-strain curves using Equation (4), at the same time points as the ones available from the experimental data.
- **Step 4**: For each time point, calculate the difference between the experimental and analytic values (identification error).
- **Step 5**: Using a Levenberg-Marquardt optimization algorithm [13], change the material's parameters in order to minimize the identification error.

As explained in Section 3, three creep tests were done for each stress value, therefore nine curves were available for compression and six for tension. The procedure described above was repeated for all possible combinations of curves available at the different stresses. For the identification of compression creep parameters, the curve were combined on sets of three (one at each stress), and for the tensile creep parameters the curves were combined in sets of two.

### 4.2. Statistical Analysis

The main idea in this paper is to show how the scatter in the tests can influence the identified material parameters, in terms of their range of variation.

First, two concepts should be defined:

- Statistical population: group of all possible items in the study domain. In the present case, the population is the infinite number of creep tests that could be done.
- Statistical sample: the actual subset of the population being studied. In this study, the statistical sample is used to draw conclusions about the statistical population, since the mean and the standard deviations of the population are unknown.

Confidence intervals can be used to predict what is the confidence level that one parameter of the statistical population (for example, the average) lies in a given range, calculated using the statistical sample.

For example, if one defines a 70% confidence interval for the average of the material parameter *A* in Equation (2), this means that the interval resulted from the estimation procedure is 70% reliable, not that there is 70% probability that the parameter *A* for the statistical population lies within this interval.

Once the inverse identifications were made using the different possible combinations of curves, the average and standard deviation were calculated.

Assuming a normal distribution for the results, confidence intervals can be calculated according to the following expression:

$$\left( \overline{x} - t^* \frac{\eta}{\sqrt{n}}, \overline{x} + t^* \frac{\eta}{\sqrt{n}} \right) \tag{6}$$

where $\overline{x}$ is the average, $\eta$ is the standard deviation of the statistical sample, $n$ is the number of observations and $t^*$ is the critical value according to Student's t-distribution, that can be found in specialized tables [14].

Once the confidence intervals were obtained, the strain vs time curves were plotted using their extreme values, to check the variation of the creep curves within the chosen confidence level.

Due to the high heterogeneity of refractories and limited amount of experimental data available, it was decided to plot 70% confidence intervals for all the analyses in Section 5.

## 5. Results and Discussions

All the curves presented in this work represent the average of the two extensometers used to measure the displacements over time, as explained in Section 3. The compression creep curves were already reported by Samadi et al. [12], where a different statistical approach was applied for the parameter determination.

Figure 3, shows the effect of the stress increase on the creep strain for the studied material. It can be observed that for a compression stress $\sigma = 8$ MPa, the time to the complete failure of the sample is approximately 11.5 h, while at $\sigma = 10$ MPa it is reduced to less than 1.5 h. The same effect can be observed in tension, where a variation from $\sigma = 0.20$ MPa to $\sigma = 0.25$ MPa resulted in a decrease in the test time from approximately 33 min to 2.5 min.

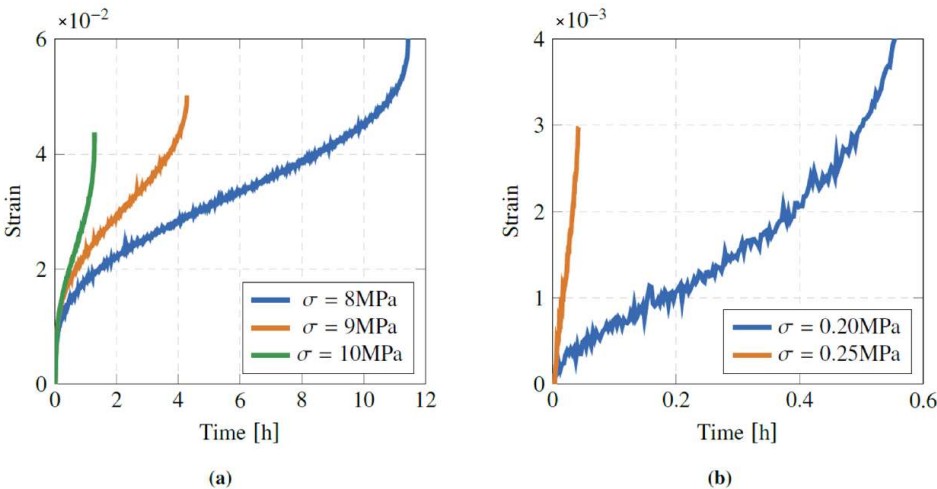

**Figure 3.** Creep tests at 1300 °C. (**a**) Compression. (**b**) Tension.

It's well known that refractories are heterogeneous materials, since normally they have large grain sizes compared to the size of the samples used for mechanical test, with some exceptions. As such, it is common to observe a considerable scatter in the data regarding their mechanical properties. Figure 4a shows that, for the material studied in this work, under a compression stress $\sigma = 8$ MPa, Sample 1 failed after 11.5 h, while Sample 3 failed after 3 h. Under a tension load of $\sigma = 0.2$ MPa, Sample 1 failed after 1.1 h, while Sample 3 failed after 30 min of test. For Sample 2 the test was interrupted after 20 min due to sudden failure of the glue, although the resulting creep curve is similar to the one of Sample 3.

Figure 5a shows all the compression creep curves obtained experimentally at the three different stress levels, but only for the primary and secondary creep stages. Although it is still possible to observe a scatter in the data, this effect is much less pronounced than when the third creep stage is also considered, like in Figure 4. More particularly, the results of Samples 3 and 6 are very similar, although the stress levels were $\sigma = 8$ MPa and $\sigma = 9$ MPa, respectively. The results of the tensile creep tests are plotted in Figure 5b.

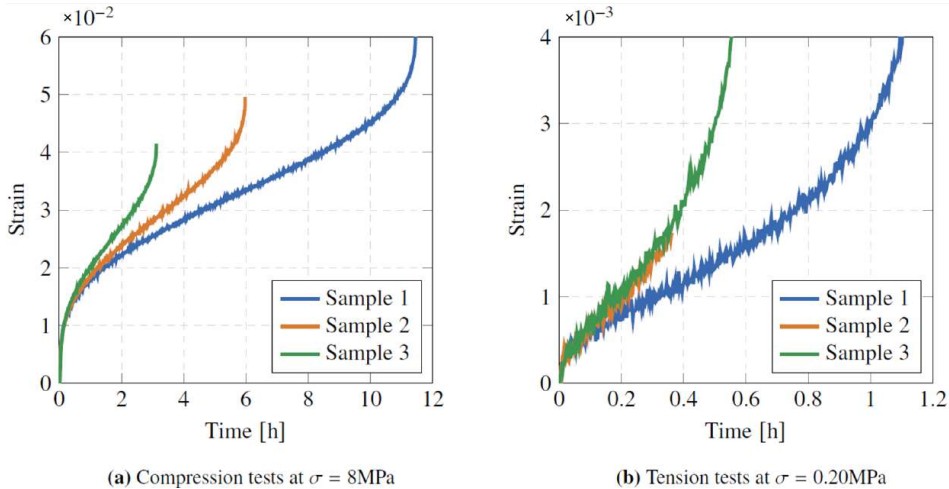

**(a)** Compression tests at $\sigma = 8$MPa

**(b)** Tension tests at $\sigma = 0.20$MPa

**Figure 4.** Three stages of creep at $T = 1300\ °$C under tensile and compressive loads.

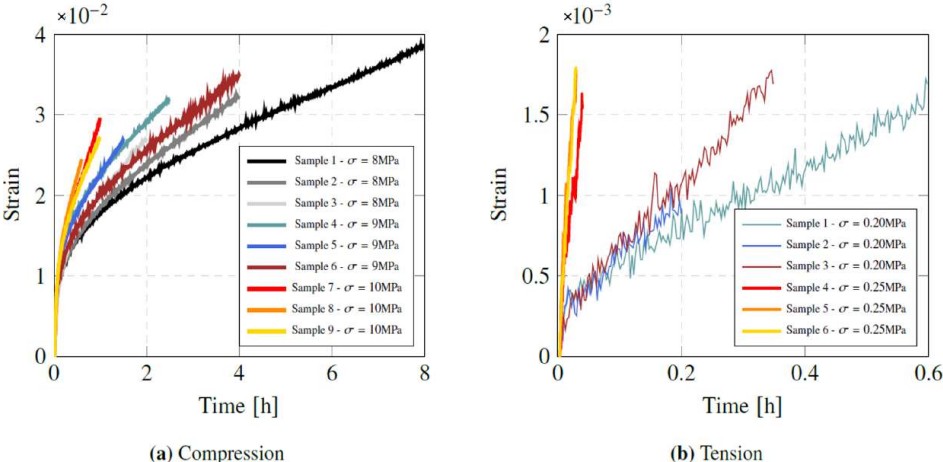

**(a)** Compression

**(b)** Tension

**Figure 5.** Creep tests at 1300 °C.

The scatter in the data, that is considered to be rather normal for refractory materials, can be explained by many factors. From the material's point of view, the heterogeneity in the bricks used to produce the samples can come from the production processes, such as the pressing and the heat treatment. From the testing procedures, micro-cracking of the sample during its production and misalignment of the load can contribute to the variations in the results. It's out of the scope of this work to precisely define the causes of this scatter, although the authors believe it can come from a combination of all factors mentioned.

*Identification of the Creep Parameters*

The curves presented in Figure 5 were used to identify the creep parameters related to Equation (2). It should be noticed that frequently the main goal of creep parameters identification for a given material is to later model more complex structures under multidimensional loads. In this case, the stage of the creep needed for simulation must be defined (primary or secondary), since Equation (2) does not provide a criteria to transit from the first to the second creep stages during the calculations.

From Figure 5a, it is observed that primary creep stage has an important influence in the time-strain response under compression, and therefore it will be considered during the identification. However, the tensile creep data presented in Figure 5b shows that, although the occurrence of primary creep stage can be observed, it finishes after a few minutes and secondary creep holds for most of the test time. For this reason, the secondary creep assumption (Equation (2) with $a = 0$) seems

to be a feasible approximation of the material behavior in tensile creep regime, and was used for the identifications.

Table 1 shows the inversely identified compressive creep parameters. The nine experimental curves were combined in sets of 3, resulting in 27 combinations. The 70% confidence interval was calculated using a Student's t-distribution critical value of $t^* = 1.057$.

In Figure 6 the creep curves resulting from the average of the identification parameters were plotted together with the experimental curves. It is possible to observe a good agreement between the experimental and identified results. The figure also shows the upper and lower bound creep curves resulted from the extreme values of the confidence intervals. It can be concluded that, to be 70% confident about the identification procedure, the possibility of a large variation of the average values must be assumed. This fact is due to the limited number of tests that could be done, considering the high cost and the time demand to perform them.

**Table 1.** Results of the inverse identification—Compression creep.

| Parameter | Average | Std. Deviation | 70% Confidence Interval |
|:---:|:---:|:---:|:---:|
| $\log_{10}(A[\text{MPa}^{-n}\text{s}^{-1}])$ | $-13.52$ | $0.925$ | $(-14.08, -12.95)$ |
| $n$ [-] | $3.56$ | $0.554$ | $(3.22, 3.90)$ |
| $a$ [-] | $-2.59$ | $0.218$ | $(-2.73, -2.46)$ |

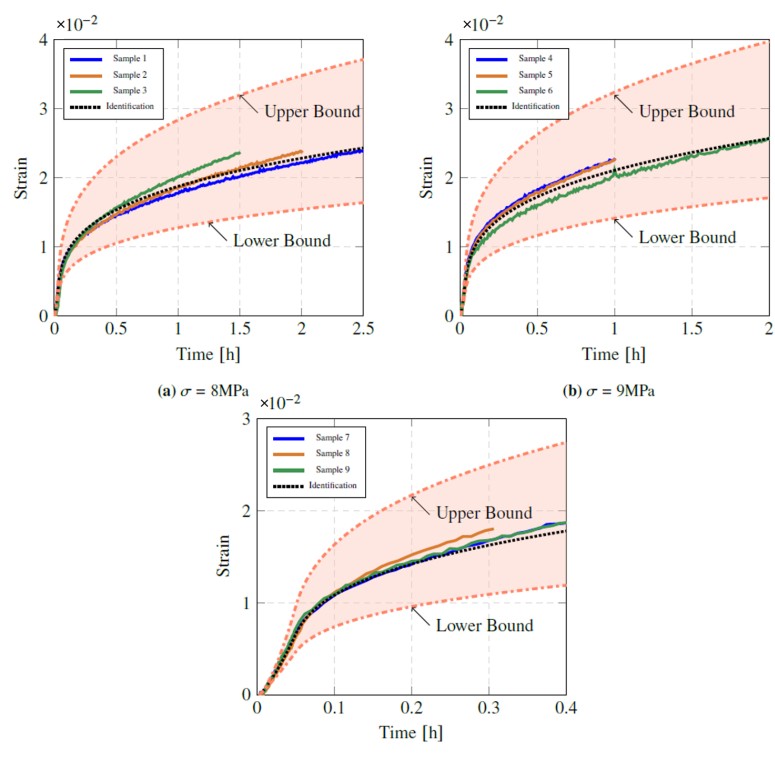

**Figure 6.** Compressive creep identification results at $T = 1300\,°\text{C}$.

Table 2 shows the identified parameters for the tensile creep and a 70% confidence interval, calculated with $t^* = 1.108$, and Figure 7 shows the experimental and identified creep curves. Such as in the compression case, the creep curve resulting from the average identification is in a good agreement with the experimental curves, but the standard deviation has a very high value compared to the average. This fact, combined with the resulted number of experimental data available, results in a broad confidence interval, and the upper and lower bounds were not plotted in Figure 7.

**Table 2.** Results of the inverse identification—Tensile creep.

| Parameter | Average | Std. Deviation | 70% Confidence Interval |
|---|---|---|---|
| $\log_{10}(A[MPa^{-n}s^{-1}])$ | 2.52 | 1.14 | $(1.62, 3.42)$ |
| $n$ [-] | 12.05 | 1.78 | $(10.65, 13.45)$ |

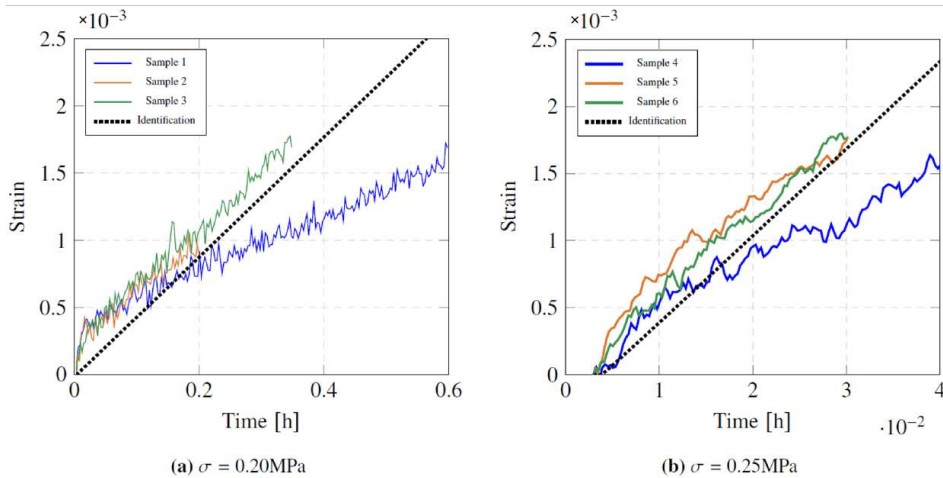

**Figure 7.** Tensile creep identification results at $T = 1300\ ^\circ C$.

Considering the possible effects the test procedures as well as the material heterogeneity, another approach for the inverse identification is to only consider the two closer curves at each load, and to eliminate the curve that deviate considerably from them. For example, considering the compressive creep curves presented in Figure 5a at $\sigma = 9$ MPa, Samples 4 and 5 are in good agreement between each other, while Sample 6 seems to deviate. In the same way, for the tensile tests at $\sigma = 0.20$ MPa, Sample 1 presents a considerable difference when compared with Samples 2 and 3.

This methodology needs to be used carefully, because is not always obvious when a deviating result comes from a problem due to the testing procedure or due to an abnormal variation of the material, and when it is due to its actual normal heterogeneity. A reliable way to make this verification is to perform a higher number of tests, what presents the difficulties already mentioned.

To apply this methodology, Samples 3, 6 and 8 were removed from the identification of compressive parameters, and Samples 1 and 4 were removed from the identification of tensile parameters.

Table 3 shows the identification results for the compressive creep tests with the reduced number of samples, and Table 4 the results of the tensile creep parameters. It can be seen that, when comparing to Tables 1 and 2, the average of the parameters changed less than 10%, but the standard deviation was reduced up to 50%. It should be noticed that, since the number of experimental curves being considered decreased, the critical value of the Student's t-distribution increased, being now $t^* = 1.134$ for compression and $t^* = 1.25$ for tension. Nevertheless, the decrease of the standard deviation was more influential than the increase of $t^*$, resulting in a more restricted range of variation for the confidence interval. Figures 8 and 9 show the plots of the results.

**Table 3.** Results of the inverse identification—Compression creep—Reduced number of samples.

| Parameter | Average | Std. Deviation | 70% Confidence Interval |
|---|---|---|---|
| $\log_{10}(A[MPa^{-n}s^{-1}])$ | $-14.16$ | 0.506 | $(-14.49, -13.83)$ |
| $n$ [-] | 3.96 | 0.257 | $(3.79, 4.13)$ |
| $a$ [-] | $-2.74$ | 0.142 | $(-2.83, -2.64)$ |

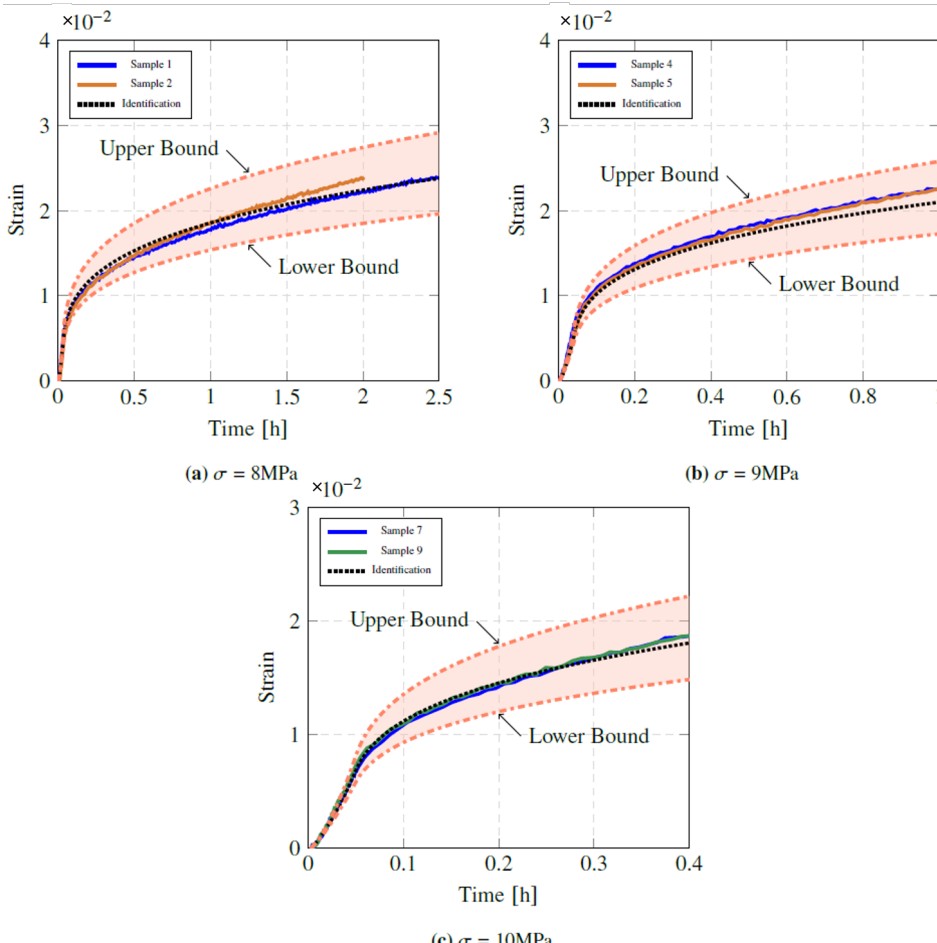

**Figure 8.** Compressive creep identification results at $T = 1300\,^\circ$C—Reduced number of samples.

**Table 4.** Results of the inverse identification—Tensile creep—Reduced number of samples.

| Parameter | Average | Std. Deviation | 70% Confidence Interval |
|---|---|---|---|
| $\log_{10}(A[\text{MPa}^{-n}\text{s}^{-1}])$ | 2.48 | 0.107 | $(2.39, 2.56)$ |
| $n$ [-] | 11.86 | 0.168 | $(11.73, 11.99)$ |

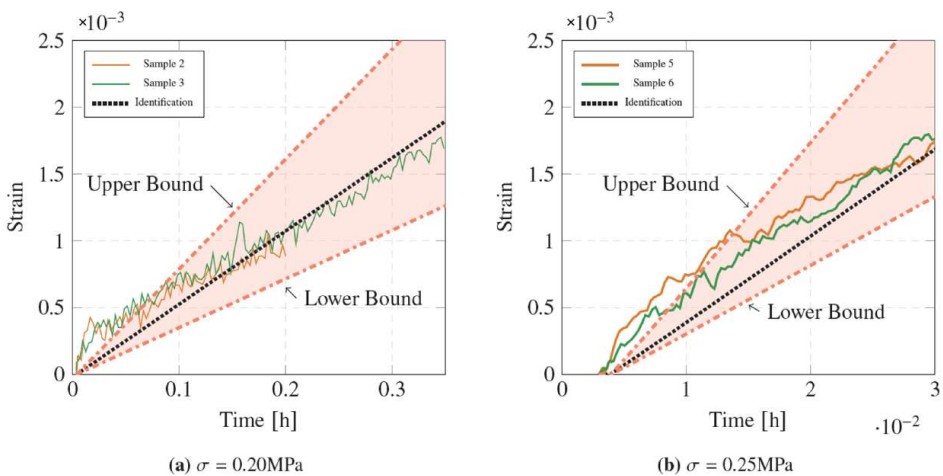

**Figure 9.** Tensile creep identification results at $T = 1300\,^\circ$C—Reduced number of samples.

## 6. Conclusions

This work presented the characterization of the tensile and compressive creep behavior of a shaped Alumina-Spinel material used in the working lining of steel ladles. Experimental curves were presented, as well as the results of an inverse identification of the parameters of a Norton-Bailey creep law.

The analytic creep curves resulting from the average of the identified parameters showed a good agreement with the experimental curves, although the variation range of the confidence intervals can be decreased if more experimental curves are available, what is not always possible due to limitations of cost and time.

The parameters identified in this work can be used in finite elements software to predict the creep strains and stress relaxations of complex refractory structures.

**Author Contributions:** Conceptualization, L.T., D.G. and E.B.; Formal analysis, L.T.; Funding acquisition, D.G. and E.B.; Investigation, L.T. and S.S.; Methodology, L.T. and S.J.; Project administration, D.G. and E.B.; Resources, J.G., D.G. and E.B.; Software, L.T.; Supervision, J.G., S.J., T.S., D.G. and E.B.; Visualization, L.T.; Writing—original draft, L.T.; Writing—review & editing, S.S., J.G., S.J., T.S., D.G. and E.B. All authors have read and agreed to the published version of the manuscript.

**Funding:** This work was supported by the funding scheme of the European Commission, Marie Skłodowska-Curie Actions Innovative Training Networks in the frame of the project ATHOR—Advanced THermomechanical multiscale mOdelling of Refractory linings 764987 Grant.

**Acknowledgments:** This work was supported by the funding scheme of the European Commission, Marie Skłodowska-Curie Actions Innovative Training Networks in the frame of the project ATHOR—Advanced THermomechanical multiscale mOdelling of Refractory linings 764987 Grant. The authors acknowledge the company RHI Magnesita for providing the materials.

**Conflicts of Interest:** The authors declare no conflict of interest. The funders had no role in the design of the study; in the collection, analyses, or interpretation of data; in the writing of the manuscript, or in the decision to publish the results.

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
