# Peer review of "Experimental Investigation of the Tension and Compression Creep Behavior of Alumina-Spinel Refractories at High Temperatures"

_ceramics, doi:10.3390/ceramics3030033_

Round 1

Reviewer 1 Report

This is a paper well presented. Due to the limited sizes of the samples tested the results are of interest and validate the methology used for the inverse identification of the parameters, in the Norton-Bailey based law.It does allow to appreciate the influence of both the time and applied stress at a given temperature. The influence of temperature would also have been of interest. 

Author Response

Dear reviewer.

Thank you very much for your comments. As for the influence of temperature, we are planning to write another publication about this soon, so we decided to focus only in the statistical analysis at a given temperature for the current paper.

Reviewer 2 Report

The paper is a study of statistical analysis of creep measurements and the influence of data scattering on the evaluation of material parameters. While I do find the research interesting and of importance to the readers, I nevertheless suggest some improvements to be made to the manuscript.

  1. Language and style: the manuscript is well written, English is good and I have no remarks in that area (not being a native English speaker myself, I also do not spot every single mistake anyway).
  2. Introduction: The introduction gives some basic knowledge about creep behaviour, which every researcher working in the field should be familiar with (Figure 1 showing the stages of creep…). If I understand correctly, the first author is a beginning stage researcher, so this is in a way understandable. However, I suggest the authors focus more on the research of relevant literature and not on the textbook information.
  3. Material and Experiments: Well written and describes the experiments performed in detail. No other comments here.
  4. Creep model and Inverse identification: Since there are no results of the work yet, in my opinion this sections need to be considered in the Introduction (and parts that are relevant to the results should be moved to results)
  5. Results and Discussion: I find this section well written, with a clear aim and good data interpretation. It is interesting and also a reminder to not trust data from a single specimen.
  6. Conclusion: no special comments.

Overall, I find the manuscript well written and informative, with clear data interpretation and analysis. I suggest the authors to slightly rearrange the flow of the manuscript and to adapt a less textbookish approach to the introduction. Regarding results and discussion section, I have no remarks to reanalyse the data, so I suggest a major revision of the manuscript.

Author Response

Dear reviewer,

Thank you very much for your valuable help. After your comments, I did the following modifications to the paper:

1- Removed the pictures showing the creep stages and the influence of temperature and time, but kept the text about this.

2- Moved the section "Creep model" to the introduction.

3- Changed the name of the section "Inverse identification" to "Methodology". This section now contains a subsection called "Inverse identification" and another called "Statistical analysis". I believe this will be better for the organization of the paper, since this is a long section and would made the introduction too long.